

# Detection of carbon monoxide pollution from megacities and wildfires on regional and urban scale: The benefit of CO column retrievals from SCIAMACHY 2.3 μm measurements under cloudy conditions

Tobias Borsdorff[1], Josip Andrasec[1], Joost aan de Brugh[1], Haili Hu[1], Ilse Aben[1], and Jochen Landgraf[1]

[1]SRON Netherlands Institute for Space Research, Utrecht, the Netherlands

*Correspondence to:* T. Borsdorff (t.borsdorff@sron.nl)

**Abstract.** In the perspective of the upcoming Sentinel-5 Precursor carbon monoxide data product, we discuss the benefit of CO total column retrievals from cloud contaminated SCIAMACHY 2.3 $\mu$m shortwave infrared spectra to detect atmospheric CO enhancements on regional and urban scales due to emissions from megacities and wildfires. The study uses the operational Sentinel-5 Precursor algorithm SICOR, which infers the vertically integrated CO column together with effective cloud parameters. We investigate the capability to detect localized CO enhancements distinguishing between clear-sky observations and observations with low and medium-high clouds. Exemplary, we analyze CO enhancements over the megacities Paris, Los Angeles, and Tehran as well as the wildfire events in Mexico/Guatemala 2005 and Alaska/Canada 2004. The CO average of the SCIAMACHY full mission data set of clear-sky observations can detect weak CO enhancements of less than 10 ppb due to air pollution in these cities. For low cloud conditions, the CO data product performs similarly well. For medium-high clouds, the observations show a reduced CO signal both over Tehran and Los Angeles, while for Paris no significant CO enhancement can be detected. This indicates that information about the vertical distribution of CO can be obtained from the SCIAMACHY measurements. Moreover, for the Mexico/Guatemala fires, the low-cloud CO data captures a strong outflow of CO over the Gulf of Mexico and the Pacific Ocean and so provides complementary information to clear-sky retrievals. For both burning events, enhanced CO values are even detectable with medium-high cloud retrievals, confirming a distinct vertical extension of the pollution. The larger number of additional measurements and hence the better spatial coverage, improves significantly the detection of wild fire pollution using both the clear-sky and cloudy CO retrievals. Due to the improved instrument performance of the TROPOMI instrument with respect to its precursor SCIAMACHY, the upcoming Sentinel-5 Precursor CO data product will allow to detect CO emission and its vertical extension of many more cities and wildfires and so opens new research opportunities.

## 1 Introduction

Carbon Monoxide (CO) is an atmospheric trace gas emitted mainly by incomplete combustion processes. Its oxidation with the hydroxyl radial (OH) represents its major sink (Spivakovsky et al., 2000). With its moderate long lifetime of several months and





its low background concentration (Holloway et al., 2000), it is an important tracer for atmospheric transport of pollution (Logan et al., 1981). From space, CO is measured by different satellite instruments with global coverage, e.g. MOPITT (Measurements of Pollution in the Troposphere; Deeter (2003)), AIRS (Atmospheric Infrared Sounder; McMillan (2005)), TES (Tropospheric Emission Spectrometer; Rinsland et al. (2006)), IASI (Infrared Atmospheric Sounding Interferometer; Turquety et al. (2004)),

and SCIAMACHY (Scanning Imaging Absorption Spectrometer for Atmospheric Chartography; Gloudemans et al. (2009); Frankenberg et al. (2005); Buchwitz et al. (2007); Gimeno Garcia et al. (2011)).

For the interpretation of satellite observations, the presence of clouds in the observed scene represents a major challenge. Here, light scattering and hence the shielding of the atmosphere below the cloud affects the vertical sensitivity of the measurement. This hampers the retrieval of the vertically integrated total column of CO from cloudy observations and different

approaches have been proposed to cope with this problem. Deeter (2003); Buchwitz et al. (2004); de Laat et al. (2006); Borsdorff et al. (2016) suggest to consider only observations under clear sky conditions or weakly cloud contaminated, assuming that the sensitivity to CO in the lower troposphere is sufficient to estimate the total CO column. Obviously, this assumption does not hold for optically thick water clouds shielding the atmosphere below. Therefore, Buchwitz et al. (2007); Gloudemans et al. (2009); de Laat et al. (2012) used the retrieved $CH_4$ total column and model vertical profiles of $CH_4$ and CO to com-

pensate for the shielding effect by clouds on the estimated total column of CO. Alternatively, Rinsland et al. (2006); Borsdorff et al. (2017); Vidot et al. (2012) discussed the retrieval of the CO column jointly with effective cloud parameters resulting in a retrieved CO column with its vertical sensitivity, where the latter reflects the effect of clouds on the light path and so includes the shielding effect of clouds. This approach is not limited to particular conditions of cloud coverage and so generalizes the above-mentioned techniques providing a higher data yield.

The usefulness of CO total column from satellite observations have been demonstrated by several studies. For example, after temporal averaging of several years of IASI and SCIAMACHY CO measurements, (Pommier et al., 2013; Buchwitz et al., 2007; Clerbaux et al., 2008) detected the relatively weak CO enhancement of urban pollution in megacities. Also, the pronounced enhancement of CO due to wildfires have been reported (e.g., Gloudemans et al., 2006; Buchwitz et al., 2007). Depending on the study, only clear-sky observations or both clear-sky and cloudy observations are used. Due to the different

vertical sensitivity of the observations, the use of the data have to be considered with care. For moderately high clouds, observations might not be suited to detect enhanced CO concentrations in the lower atmosphere because of the shielding of the atmosphere below the cloud. On the other hand, using cloudy observations jointly with clear sky may provide new applications to constrain the vertical extension of the pollution (Liu et al., 2014). This aspect is particularly interesting in the light of upcoming future missions with improved radiometric and spatial sampling performance.

October 13, 2017 the Tropospheric Monitoring Instrument (TROPOMI) was successfully launched on the Sentinel-5 Precursor Mission (S-5P) (Veefkind et al., 2012). It measures Earth reflected radiances in the ultraviolet, visible, near infrared and shortwave infrared spectral range with a spatial resolution of about $7 \times 7$ km$^2$ at sub-satellite point and daily global coverage. Here, shortwave infrared (SWIR) observations in the 2.3 $\mu$m spectral region provide information on the CO total column amount. In recent years, the Shortwave Infrared CO Retrieval algorithm SICOR has been developed for the operational

processing of TROPOMI data (Vidot et al., 2012; Landgraf et al., 2016b, a). TROPOMI's high signal-to-noise ratio (SNR)



performance in the SWIR will provide clear sky CO total column densities with vertical sensitivity throughout the atmosphere (e.g., Buchwitz et al., 2004; Gloudemans et al., 2008; Borsdorff et al., 2014) and with a precision < 10 % for single clear sky soundings (Landgraf et al., 2016b). However even with the high spatial resolution and sampling of TROPOMI, a major part of the measurements is cloud contaminated (Krijger et al., 2005, 2011). To optimally explore the SWIR measurements, SICOR

retrieves CO also for cloudy conditions over land and ocean inferring effective cloud parameters (cloud optical thickness $\tau_{cld}$, cloud centre height $z_{cld}$) together with trace gas columns (Vidot et al., 2012; Landgraf et al., 2016b). The TROPOMI CO data product comprises the estimate of the CO column, its noise estimate, effective cloud parameters and the CO column averaging kernel, which reflects the effect of cloud shielding and light path enhancement by clouds on the retrieved CO column. Furthermore, the effective cloud parameters provide useful information to e.g. classify measurements by the type of cloud

contamination.

To evaluate the benefit and maturity of this approach for the detection of localized CO enhancements by TROPOMI we have applied the SICOR algorithm to the full SCIAMACHY mission data set of 2.3 $\mu$m spectra. Here, SCIAMACHY covers the TROPOMI SWIR band with the same spectral resolution but with inferior radiometric performance, spatial resolution and global coverage (Bovensmann et al., 1999). The SCIAMACHY CO dataset was validated with TCCON/NDACC measurements

for clear-sky observations over land (Borsdorff et al., 2016) and with TCCON and MOSAIC/IAGOS airborne measurements (Borsdorff et al., 2017) for cloud contaminated measurements over land and oceans. In this study, we discuss the benefit of CO retrievals from cloud contaminated SCIAMACHY 2.3 $\mu$m measurements with their intrinsic vertical sensitivity for the detection of CO pollution from megacities and wildfires. Here, we distinguish CO retrievals from measurements under clear-sky, low and medium-high cloud conditions. Exemplary, we discuss the CO pollution by Tehran, Paris, and Los Angeles as

well as the wildfires in Mexico/Guatemala 2005 and Alaska/Canada 2014.

The paper is structured as follows: In section 2, we present the SCIAMACHY CO dataset. Section 3 analyzes the benefit of the SCIAMACHY CO retrievals under cloudy conditions to detect wildfires, and Section 4 focuses on the CO emission from megacities. Finally, Section 6 will give a summary and conclusions.

## 2   SCIAMACHY CO dataset

In this study, we analyze the SICOR CO total column densities, retrieved from individual SCIAMACHY 2.3$\mu$m spectra for the entire period of the mission from January 2003 until April 2012. The CO data product is available on the public ftp site ftp://ftp.sron.nl/pub/pub/DataProducts/SCIAMACHY_CO/. It consists of the estimates of the total column concentrations of CO, $H_2O$ and HDO ($c_{CO}$, $c_{H2O}$, $c_{HDO}$), the corresponding retrieval noise ($\epsilon_{CO}$, $\epsilon_{H2O}$, $\epsilon_{HDO}$), averaging kernels, effective cloud parameters (cloud optical thickness $\tau_{cld}$ and cloud height $z_{cld}$) and the SWIR Lambertian surface albedo. Moreover,

different auxiliary parameters are provided like the number of iterations of the inversion ($N_{iter}$) and the maximum signal to noise ratio of the measurement $SNR_{max}$.

SICOR uses the profile scaling approach, discussed in detail by Vidot et al. (2012); Borsdorff et al. (2014); Landgraf et al. (2016b). Here, the effective cloud parameters are estimated using prior knowledge about $CH_4$ using the ECMWF surface





pressure and the corresponding absorption in the spectral fit window. The algorithm can also be used for SCIAMACHY
CO data processing because of the similarity of the TROPOMI and SCIAMACHY observations. The specific SCIAMACHY
settings, e.g. the selection of the retrieval window, are discussed by Borsdorff et al. (2017). Due to an ice layer on the SWIR
detectors and the radiometric degradation of the instrument, the processing of SCIAMACHY CO data requires a radiometric

re-calibration of the SWIR spectra as described by Borsdorff et al. (2016).

In this study, we apply a data screening of the individual SCIAMACHY CO retrievals based on the number of iterations
$N_{\text{iter}}$ and the estimate of the retrieval noise, which we compared with $\sigma$ the difference of the 50th and 68th percentile of the
retrieval results for the different data ensembles. The data filter reads

    1. $N_{\text{iter}} < 15$

2. $\epsilon_{CO} < 4.5\,\sigma_{CO}$

    3. $\epsilon_{H2O} < 4.5\,\sigma_{H2O}$

    4. $\epsilon_{HDO} < 4.5\,\sigma_{HDO}$.

For the analysis of air pollution from megacities, we add an additional filter considering the median CO column $\mu_{\text{CO}}$ of the
CO data set of the different cities,

5. $\mu_{\text{CO}} - 4.5\,\sigma_{CO} \leq c_{\text{CO}} < \mu_{\text{CO}} + 4.5\,\sigma_{CO}$ .

This filter removes outliers of our data sets, which we attribute to erroneous retrievals rather than an atmospheric signal in case
of the selected megacities. It enables us to detect the relatively weak CO enhancement after averaging the data over the entire
mission period. To distinguish the effect of clouds on the retrieval, we consider three different categories of cloudy observations
as indicated in Tab. 1.

An important element of the CO data product is the column averaging kernel $A$, which provides the sensitivity of the
retrieved CO column to changes in the true vertical profile $\rho_{true}$ of CO (Rodgers, 2000), namely

$$c_{ret} = A\rho_{true} + \epsilon_{CO} , \qquad\qquad\qquad (1)$$

where $\epsilon_{CO}$ represents the error of the retrieved CO column caused by measurement errors. Eq. (1) can be interpreted as a
weighted altitude integration accounting for the vertical sensitivity of the retrieval to estimate the retrieved CO column density.

Figure 1 shows the total column averaging kernels for four different cloud conditions over Paris. Here, scenes contaminated
by optically thin low clouds provide a good vertical sensitivity of the total column of CO and so the values of $A$ are close
to 1 for all altitudes. However, for scenes with optically thick clouds, the retrieval loses CO sensitivity below the cloud with
averaging kernel values well below 1. Because the CO column is estimated by a scaling of a reference profile, CO variations
above a cloud also induce an adjustment of the CO concentration below the cloud, where the measurement is not sensitive

to. This explains the column averaging kernel values > 1 at this altitude range (Borsdorff et al., 2016). This limited retrieval
sensitivity to the atmospheric composition below a cloud induces the so-called null-space error to the retrieved total column





(e.g., Borsdorff et al., 2014). For the profile scaling approach the magnitude of the null-space error depends on the one hand on the loss of vertical sensitivity and on the other hand on the discrepancy between the true vertical profile and the reference profile to be scaled by the inversion.

For individual CO retrievals from SCIAMACHY observations, the retrieval noise $\epsilon_{CO}$ can be high and can even exceed 100 % of the retrieved column depending on the SNR of the measurement (Gloudemans et al., 2008). Hence, for most applications individual SCIAMACHY CO retrievals need to be averaged to reduce the noise (de Laat et al., 2007; Gloudemans et al., 2006). In this study, we use an oversampling technique similar to the one used Fioletov et al. (2011). This means that we first define an equidistant latitude/longitude grid with a sampling distance $\delta$ for a considered scene. For each grid cell, an averaged CO value is calculated using SCIAMACHY CO retrieval weighted with its noise error $\epsilon_{CO}$ within a circular domain of a radius $r$ around the cell centre. Here $\delta < r$, which corresponds to an oversampling of the averaged SCIAMACHY CO field.

## 3 CO pollution from wildfires

After carefully evaluating the SCIAMACHY CO data set, we have selected two examples of wildfire events for further discussion: agricultural fires in Mexico/Guatemala 2005 and forest fires in Alaska/Canada 2004. Buchwitz et al. (2007) discussed the fires in Alaska/Canada 2004 with SCIAMACHY CO retrievals and Pfister et al. (2005) quantified their CO emissions using MOPITT CO data. We will revisit those fires from the perspective of CO retrievals under cloudy conditions.

Figure 2 shows time series of individual SCIAMACHY CO retrievals over Mexico for clear-sky, low cloud, and medium-high cloud conditions as well as the daily GFED4 Burned Area product of MODIS (Randerson et al., 2017). The two burning events indicated by the GFED4 Burned Area product in 2003 and 2005 are clearly reflected in the time series of low cloud and medium-high cloud retrievals but shifted by about 45 days for both events. We ascribe this temporal shift to the atmospheric response time to built up the high atmospheric CO concentration. It is interesting to note that the retrieval shows both a slowly varying increase and decrease of the burning activities over the month and enhanced peak events, both also evident in the GFED4 Burned Area data. The time series of the clear sky data is very noisy and has significant gaps because of the dark ocean surface in the SWIR which does not permit a CO retrieval. Both hamper the detection of fire events. Nonetheless, it seems that the two fire events are also visible in the clear sky data.

From the clear-sky, low-cloud, and medium-high cloud time series we calculated daily mean values and investigated the correlation of the data sets. For the correlation between the low-cloud and clear-sky data product, the Pearson coefficient is 0.6 with a mean bias of 1.7 ppb, and a standard deviation of the differences of 32.7 ppb. The large standard deviation reflects the noise of the clear-sky data. For the Mexico region, the land surface reflectivity and so the corresponding SNR of the measurement is low causing the high retrieval noise for clear-sky cases. However, the good correlation coefficient and the low bias shows that within the noise limitation the clear-sky retrievals are in good agreement with the cloudy retrievals.

The situation differs when inspecting the CO time series for SCIAMACHY observations with low and medium-high cloud coverage. Here the data are much less noisy. In the SWIR, clouds are highly reflective, as demonstrated by Borsdorff et al. (2017) using SCIAMACHY SWIR observations, and so the improved SNR of the SCIAMACHY measurements causes a re-





duced noise in the CO data product. When correlating the low-cloud and the high-cloud retrievals, we find a Pearson correlation coefficient of 0.8, a bias of 4.6 ppb, and a standard deviation of the differences of 14 ppb. This supports our finding that both low-cloud and medium-high cloud retrievals can capture the burning events equally well, something one may expect since CO pollution from wild fires constitutes a strong source that can reach the free troposphere (see e.g. Yurganov et al. (2005)).

Figure 3 shows the spatial distribution of the SCIAMACHY CO total column over Mexico for the period 15th March - 15th May 2005 and the corresponding GFED4 Burned Area product. The SCIAMACHY data are averaged over an area with a radius $r = 90$ km and subsequently oversampled with a longitude/latitude sampling distance of $\delta = 0.5$ degree. We used the same latitude/longitude grid for the MODIS data where we sum up the burned areas for the individual grid cells. Here, the clear-sky SCIAMACHY CO data clearly show the burning hot spot around the state Yucatan Peninsula in Mexico, also indicated

by the MODIS Burned Area product. For low cloud conditions, the retrieval provides additional information showing the transport of air with high CO concentration into the Gulf of Mexico and over the Pacific Ocean, in agreement with the smoke detection of the MODIS Aqua instrument (https://earthobservatory.nasa.gov/NaturalHazards/view.php?id=14748). Also, the earlier burning event in Mexico 2003 in Fig. 2 followed a similar transport pattern of enhanced CO over the oceans (not shown). The CO observations with medium-high cloud observations still reflect the CO enhancements but the measurement

density is too low to fully capture the event. Analogously, Figure 4 shows the forest fires in Alaska/Canada from 1st July to 1st August 2004 using the same oversampled approach as in Figure 3. Because of different meteorology, clear-sky observations are less frequently for the Alaska fires than for fires in Mexico and hence clear sky and low cloud observations do not fully capture the Alaska 2004 fire event. For medium-high clouds, the corresponding CO product shows much better coverage and so can detect the wildfires in agreement with the MODIS burned area product.

The benefit of cloudy observations to detect wildfires becomes also clear when comparing the number of individual SCIA-MACHY CO sounding in Fig. 5. For the Mexico fires, the 2402 individual clear sky soundings are more than doubled (6126 soundings) when we consider low cloud observations. Additionally 3225 soundings are found with medium-high clouds. For the Alaska fires, the relative distribution changes due to the different meteorological situation but confirms a significant gain in number of observations when including cloudy measurements. Here, we obtain 1473 clear sky soundings, 2454 low cloud

and 4819 medium-high cloud soundings. Due to this, the means to observe wildfire with SCIAMACHY is clearly improved by using cloudy observations in addition to clear sky observations.

## 4   CO pollution from megacities

In this section, we selected the three megacities Paris, Tehran, and Los Angeles to discuss the relevance of cloudy observations for the detection of urban pollution. Accumulating all SCIAMACHY observations from 2003 to April 2012 around these cities

and distinguishing between clear-sky, low-cloud and medium-high cloud retrievals, we apply the oversampling approach with a longitude/latitude grid of $\delta = 0.05$ degree and an averaging radius $r = 40$ km as shown in the Figs. 6, 7, and 8 together with the MODIS urban area contours of Schneider et al. (2009). Subsequently, we calculated the CO enhancement for the three cities with respect to the background signal estimating the difference of the median CO concentration inside and outside the



urban area contours. Obviously, the separation of urban and background CO concentrations cannot fully succeed due to the SCIAMACHY pixel size, the averaging approach and atmospheric transport, however the values presented in Fig. 10 give a first indication of CO enhanced due to urban population.

For all three cities, we find that the CO enhancements of clear-sky observations coincide with the MODIS urban area
contours. Furthermore, in case of Paris we can detect enhanced CO levels over the neighboring city Rouen (see Fig. 6). The strongest CO enhancement under clear-sky condition occurs for Tehran with 8.1 ppb, closely followed by Los Angeles with 6.3, and the weakest enhancement we observe for Paris with 4.3 ppb. This difference can be explained by the different source strength but is also influenced by the measurement statistics. The detection of urban CO concentration under low cloud conditions perform comparably well with an enhancement of 8.8 ppb for Tehran, 8.3 ppb for Los Angeles and 3.4 ppb for
Paris, where for Tehran and Los Angeles the spatial distribution of the enhancements agree even better with the urban area contours. For observations with medium-high clouds, we see a less distinct CO enhancement over the three cities with 7.0 ppb for Tehran, 3.6 ppb for Los Angles and only 1.8 ppb for Paris. Medium-high clouds shield the atmosphere below and so the retrieval is less sensitive to the city pollution estimating the CO column from the measurement sensitivity above the cloud as already indicated by the column averaging kernels in Fig. 1. Consequently, measurements contaminated by medium-high
clouds in combination with clear-sky and low cloud retrievals can reveal information about the strength and vertical extension of the CO pollution.

Furthermore, including the cloudy retrievals improves the measurement statistics as indicated in Fig. 9. Including cloud contaminated soundings means about double the amount of data is available for Tehran (a factor of 2.1) and Paris (a factor of 2.6) and Los Angeles (a factor of 1.8). The relative amount of cloud contaminated measurements differs significantly per
city. For Tehran, we obtain 47 % (2674) clear-sky and 44 % (2501) and 9 % (537) low and medium-high cloud observations, respectively. A similar distribution holds for Los Angeles with 55 % (2557) clear-sky and 35 % (1630) and 10 % (482) soundings for low and medium cloud conditions, whereas for Paris the situation differs with 38 % (1338) clear-sky soundings and 22 % (766) and 40 % (1388) low and medium cloud soundings. Overall, we conclude that for the SCIAMACHY mission, cloud contaminated measurements provide valuable and complementary information to clear-sky measurements.

## 5   Implications for TROPOMI

The TROPOMI instrument was successfully launched on the ESA's Sentinel-5 Precursor mission on October 13, 2017. The 2.3 $\mu$m spectral range of TROPOMI is covered by SCIAMACHY with the same spectral resolution and spectral coverage but TROPOMI is characterized by a significantly improved SNR of the measurements, a higher spatial resolution of up to $7 \times 7\,\mathrm{km}^2$, and improved spatial sampling with daily global coverage. The spectral analogy of TROPOMI and SCIAMACHY allowed us
to apply the operational TROPOMI CO retrieval algorithm SICOR on the SCIAMACHY spectra to test its performance for cloud contaminated measurements in preparation of TROPOMI data exploitation.

The spatial sampling of continuous TROPOMI nadir SWIR measurements with a swath of 2600 km provides 300 times more soundings compared to the limb-nadir observations of SCIAMACHY with a ground pixel size of 120x30 km and a swath



of 960 km. Due to the higher SNR of TROPOMI SWIR measurements, CO total column will be provided with a precision < 10 % (Landgraf et al., 2016a, b) compared to the SCIAMACHY CO column precision of 100 % and larger (Gloudemans et al., 2008). Also the radiometric accuracy is significantly improved leading to a overall bias estimate of the TROPOMI CO columns < 10 % for clear sky and cloudy observations (Landgraf et al., 2016a, b). Therefore, TROPOMI SWIR measurements

will capture burning events, with atmospheric CO signatures significantly weaker than investigated in this study, and urban pollution on a day-to-day basis with high spatial resolution without precedent. Here, using cloudy data complementary to clear-sky observations will give us a new opportunity to study the vertical and horizontal distribution of atmospheric CO pollution.

## 6 Summary and Conclusions

In this study, we discussed the benefit of using CO total column retrieval from cloud contaminated SCIAMACHY 2.3 $\mu$m shortwave infrared spectra to study pollution from megacities and wild fires complimentary to clear-sky soundings. For this purpose, we applied the SICOR algorithm to SCIAMACHY observations. SICOR is developed for the operational processing of the shortwave infrared measurements of the TROPOMI instrument on ESA's Sentinel-5 Precursor. It provides the possibility to retrieve effective cloud parameters together with trace gas columns. To investigate the capability to detect localized CO

enhancements at urban areas and wild fires, we distinguished between retrievals under clear-sky, low cloud and medium-high cloud atmospheric conditions. As an example, we analyzed CO enhancements over the megacities Paris, Los Angeles, and Tehran as well as the wildfire events in Mexico/Guatemala 2005 and Alaska/Canada 2004.

After data averaging over the entire mission period, we found that SCIAMACHY clear-sky observations can detect weak CO enhancements of less than 10 ppb over the three considered megacities and coincide with the MODIS urban area contours.

For Paris, it was even possible to detect pollution over the neighboring city of Rouen. Furthermore, clear-sky retrievals turned out to be suitable to locate the source of biomass burning in Mexico/Guatemala in agreement with the daily GFED4 Burned Area data product. Here, the sensitivity of SWIR measurements to CO throughout the atmosphere including the planetary boundary layer makes clear-sky retrievals a preferable choice for the detection of such sources. However, only a fraction of all measurements fall into this category. For example, due to the meteorological situation during the Alaska/Canada 2004 burning

event, insufficient clear sky measurements were available to fully capture the wild fires. Moreover, the noise of the retrievals strongly depends on the surface reflectivity. We found clear-sky retrievals for the wild fires in Mexico/Guatemala 2005 inferior to cloudy retrievals regarding the noise performance and temporal and spatial sampling.

Considering pollutions from megacities, the CO retrieval performs equally well for clear sky and low cloud measurements. For Tehran and Los Angeles the temporal and spatial sampling of low cloud observations improved the spatial coincidence of

the CO enhancement and the MODIS urban area product compared to the clear sky CO product. The low cloud retrievals of the 2005 wild fires in Mexico/Guatemala provides complementary information compared to clear-sky retrievals indicating the CO outflow over the Gulf of Mexico and the Pacific Ocean, which is confirmed by smoke observations of the MODIS/Aqua



instrument. Here, the high reflectivity of clouds allows the retrieval of CO over oceans, which was not possible with clear-sky measurements due to the dark ocean surface in the shortwave infrared spectral range.

The analysis of the CO pollution from megacities from SCIAMACHY soundings with medium-high clouds indicated a significant reduction in the CO enhancement for the three cities. Here, clouds shield the CO pollution and consequently, the retrieval underestimates the total column of CO. This effect differs for the three cities. While the pollution from Tehran and Los Angles are still present in the data product for medium-high clouds, it nearly vanishes for Paris, pointing to a CO enhancement localized in the lowest altitude range. Comparing low and medium-high cloud conditions for the Mexico fires, the CO enhancement is detected equally well, which indicates that the CO emission by this strong burning reaches the free troposphere. These examples show that CO retrievals for different cloud conditions are valuable to gain information about the vertical extent of the atmospheric CO pollution.

Overall, the study of SCIAMACHY CO retrievals from cloud contaminated 2.3 $\mu$m measurements showed the additive value of the data product compared to clear sky retrievals to study CO pollution on regional and urban scales. Particularly in perspective of the upcoming Sentinel-5 Precursor mission with the TROPOMI instrument as its single payload, the corresponding CO data product will open op new research opportunities due to the groundbreaking capabilities of the TROPOMI instrument.

# 7 Data availability

The full-mission SCIAMACHY CO data set used in this study, including clear-sky and cloudy-sky observations is available for download at ftp://ftp.sron.nl/pub/pub/DataProducts/SCIAMACHY_CO/. The underlying data of the figures presented in this publication can be found at ftp://ftp.sron.nl/open-access-data/.

*Acknowledgements.* SCIAMACHY is a joint project of the German Space Agency DLR and the Dutch Space Agency NSO with contribution of the Belgian Space Agency. This research has been funded in part by the TROPOMI national program from the Netherlands Space Office (NSO). Simulations were carried out on the Dutch national e-infrastructure with the support of SURF Cooperative.




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





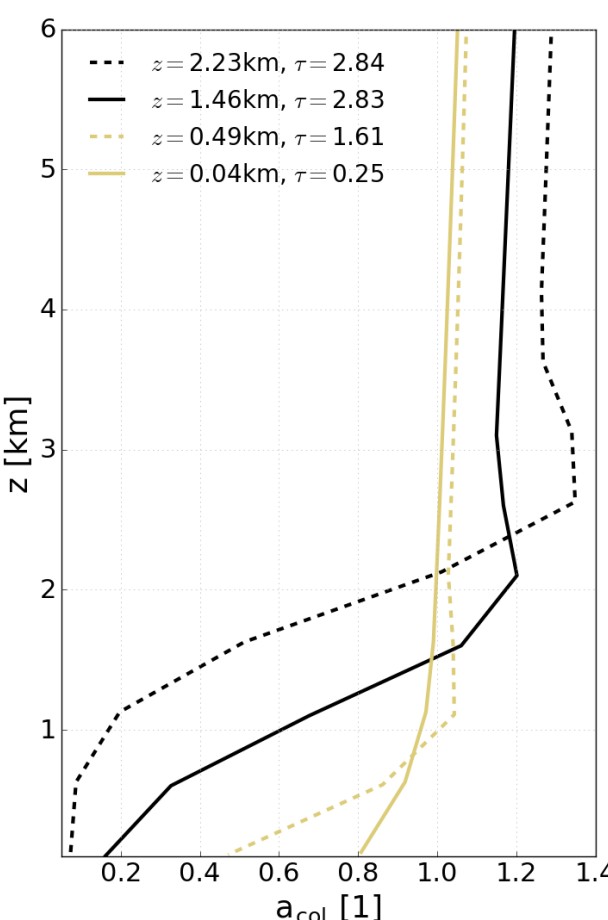

**Figure 1.** SCIAMACHY CO total column averaging kernels for different cloud centre heights ($z_{cld}$) and cloud optical thicknesses ($\tau_{cld}$). The figure shows typical cases of the vertical retrieval sensitivity of the SCIAMACHY CO retrievals over Paris.



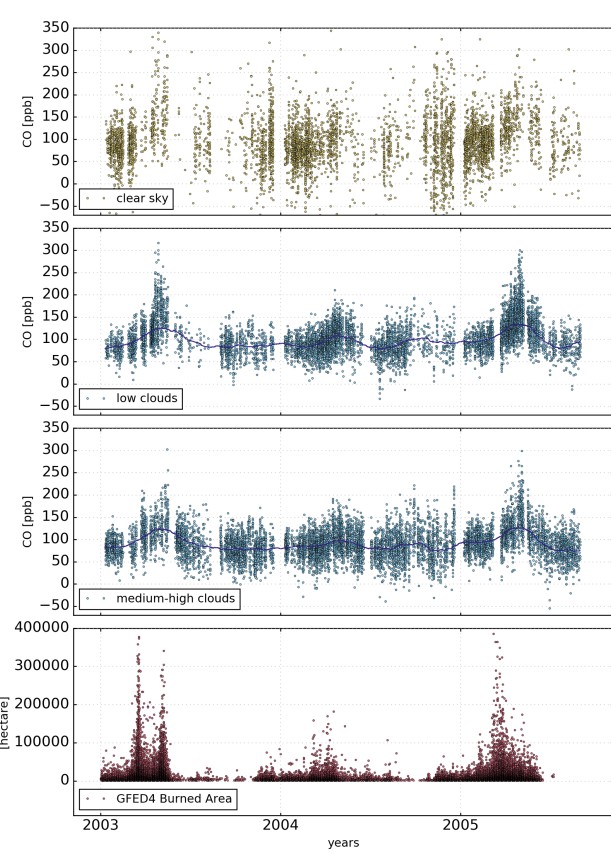

**Figure 2.** Individual SCIAMACHY CO retrievals under clear-sky, low cloud, and medium-high cloud atmospheric conditions as well as daily GFED4 Burned Area over Mexico/Guatemala in the latitude/longitude box [(22.5°N,100.0°W), (10.0°N,80.0°W)]. The blue line is a running median with half width of 30 days.





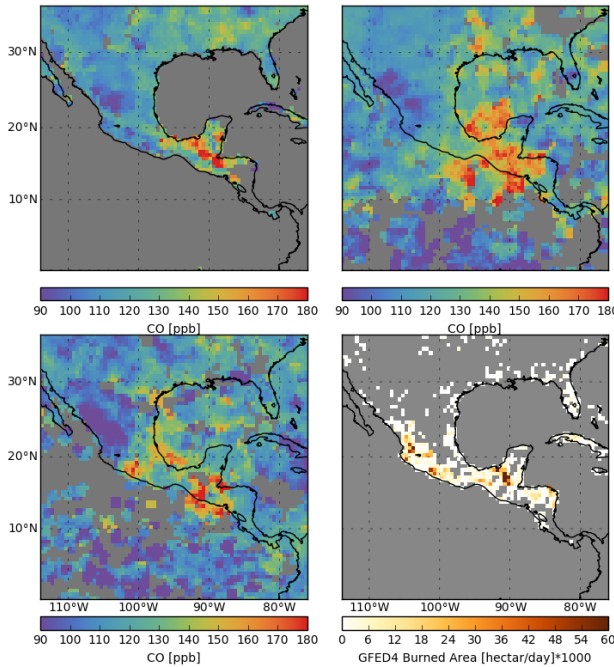

**Figure 3.** SCIAMACHY CO retrievals under clear-sky (top left), low cloud (top right), medium-high cloud (bottom left) atmospheric condition as well as daily GFED4 Burned Area (bottom right) averaged from 15th March to 15th May 2005 over Mexico and Central America in the latitude/longitude box [(36.0°N,113.9°W), (0.0°N,76.1°W)]. The resolution of the plot is 0.5 degree in latitude and longitude and the data is oversampled using a radius of 90km.

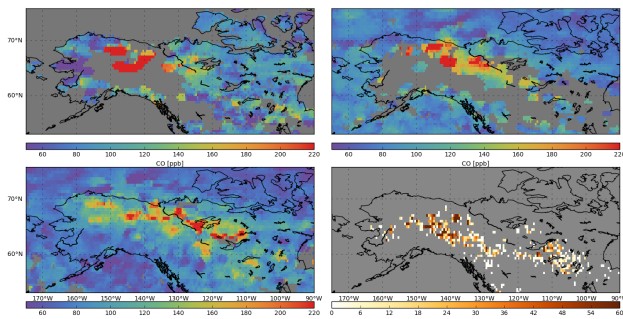

**Figure 4.** Same as Fig. 3 but for Alaska/Canada. The data is averaged from 1st July to 1st August 2004 in the latitude/longitude box [(75.0°N,175.0°W), (52.0°N,90.0°W)].



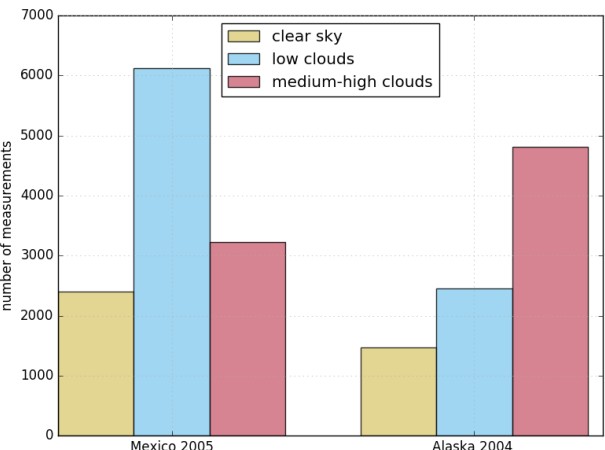

**Figure 5.** Number of individual SCIAMACHY CO retrievals under clear-sky (yellow), low cloud (blue), and medium-high cloud (pink) atmospheric conditions for the time range and latitude/longitude box specified in Fig. 3 and Fig. 4.

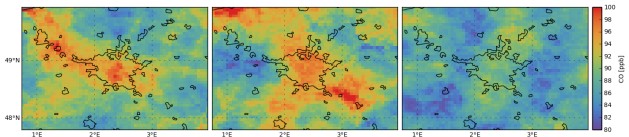

**Figure 6.** SCIAMACHY CO column mixing ratio averaged from January 2003 to April 2012 in the latitude/longitude box [(49.9°N,0.7°E), (47.8°N,4.0°E)]. The resolution of the plot is 0.05 degree in latitude and longitude and the data is oversampled using a radius of 40km.

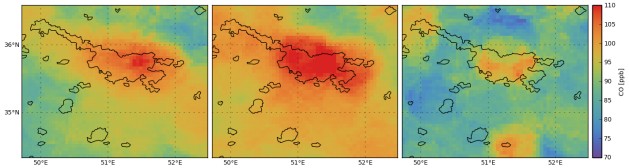

**Figure 7.** Same as Fig. 6 but for Tehran using the latitude/longitude box [(36.6°N,49.7°E), (34.3°N,52.5°E)].

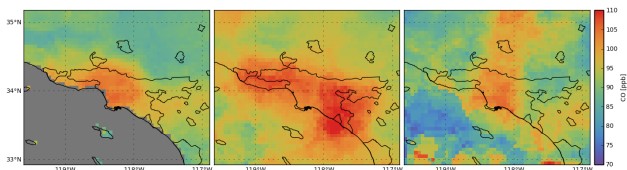

**Figure 8.** Same as Fig. 6 but for Los Angeles using the latitude/longitude box [(35.2°N,119.6°W), (32.9°N,116.9°W)].





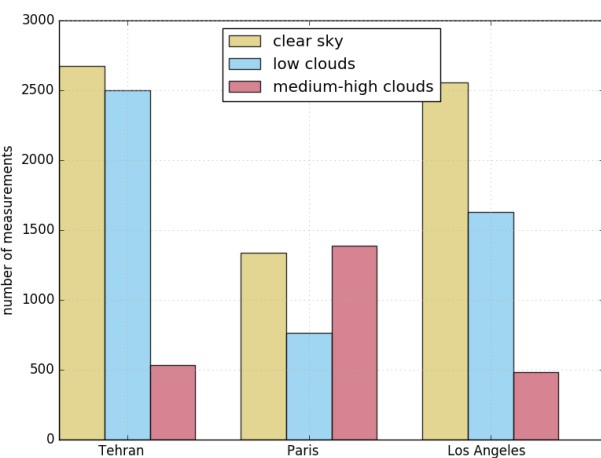

**Figure 9.** Number of individual SCIAMACHY CO retrievals under clear-sky (yellow), low cloud (blue), and medium-high cloud (pink) atmospheric condition for the time range and latitude/longitude box specified in Fig. 6, 7, and 8.

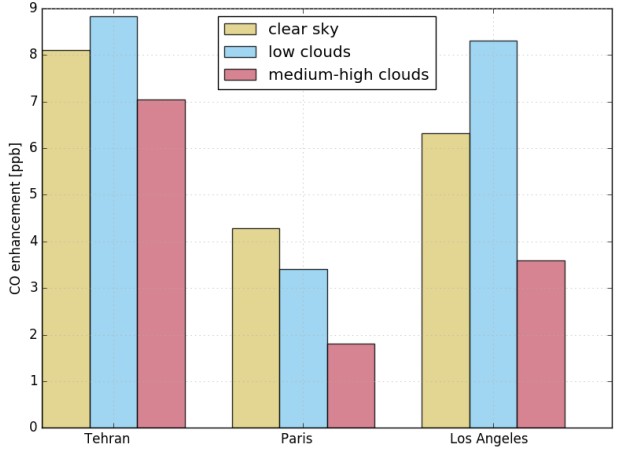

**Figure 10.** CO enhancement over cities shown in Fig. 6, 7, and 8 relative to the background concentration under clear-sky (yellow), low cloud (blue), and medium-high cloud (pink) atmospheric conditions. The difference of the median CO concentration inside and outside the urban area contours of each city is shown.





| category | optical depth | cloud height | SNR |
|---|---|---|---|
| clear-sky observations | $\tau_{cld}<2$ | $z_{cld} < 0.5\text{km}$ | $\text{SNR}_{\max} > 15$ |
| observations with low clouds | $\tau_{cld} >2$ | $z_{cld}<1.5\text{km}$ | $\text{SNR}_{\max} > 100$ |
| observations with medium-high cloud | $\tau_{cld} >2$ | $1.5\text{km}<z_{cld} < 5\text{km}$ | $\text{SNR}_{\max} > 100.$ |

**Table 1.** Categories of cloudy observations defined by the retrieved cloud optical depth $\tau_{cld}$, the cloud height $z_{cld}$ and the spectral maximum of the measurement SNR.