# Peer review of "Detection of carbon monoxide pollution from cities and wildfires on regional and urban scale: The benefit of CO column retrievals from SCIAMACHY 2.3 µm measurements under cloudy conditions"

_Atmospheric Measurement Techniques, 2017_

## Referee Comment (RC1) · Anonymous Referee #1 · 20 Feb 2018

Comments on "Detection of carbon monoxide pollution from megacities and wildfires on regional and urban scale: The benefit of CO column retrievals from SCIAMACHY 2.3 $\mu$m measurements under cloudy conditions"

Generally, the paper discusses an important subject as we lose high percent of data due to cloud screening. However, the paper needs to be more clear as it is ambiguous at many places and also it might be good if the English is edited by professionals.

[Figure]

The following are examples of the parts that need to be clarified:

1-The authors need to explain why they chose the following oversampling size?

"The SCIAMACHY data are averaged over an area with a radius r = 90 km and subsequently oversampled with a longitude/latitude sampling distance of $\delta$ = 0.5 degree"

2- The following parts need to be rephrased to be more clear

Page 2, line 13

Please rephrase to be more clear

"Obviously, this assumption does not hold for optically thick water clouds shielding the atmosphere below. Therefore, Buchwitz et al. (2007); Gloudemans et al. (2009); de Laat et al. (2012) used the retrieved CH4 total column and model vertical profiles of CH4 and CO to compensate for the shielding effect by clouds on the estimated total column of CO."

Page 2, line, 29

Please rephrase to be more clear

"This aspect is particularly interesting in the light of upcoming future missions with improved radiometric and spatial sampling performance."

Page 2, line 30 I think the following reference needs to be updated because it is 2012, and TROOMI launched in 2017.

"October 13, 2017 the Tropospheric Monitoring Instrument (TROPOMI) was successfully launched on the Sentinel-5 Precursor Mission (S-5P) (Veefkind et al., 2012). "?

Page 3, line 8 Please rephrase to be more clear

"light path enhancement"

Page 9, line 29 Please rephrase to be more clear

The following have repeated statements, please delete one of them. "a higher spatial resolution of up to 7×7 km2 , and improved spatial sampling"

Page 8, line, 28 Please rephrase to be more clear

"For Tehran and Los Angeles the temporal and spatial sampling of low cloud observations improved the spatial coincidence of the CO enhancement and the MODIS urban area product compared to the clear sky CO product."

Page 9, line, 3 Please rephrase to be more clear

"The analysis of the CO pollution from megacities from SCIAMACHY soundings with medium-high clouds indicated a significant reduction in the CO enhancement for the three cities."

Page 9, line 14 "open op new", replace op with up

The following figures have very small font so please increase their size:

Figures 2,4,,6,7,8

---

## Referee Comment (RC2) · Anonymous Referee #2 · 22 Feb 2018

Review of "Detection of carbon monoxide pollution from megacities and wildfires on regional and urban scale: The benefit of CO column retrievals from SCIAMACHY 2.3 $\mu$m measurements under cloudy conditions" (Borsdoff et al.)

This manuscript presents results showing that SCIAMACHY CO column retrievals performed under clear and cloudy conditions provide valuable, complementary information regarding both CO concentrations and height reached by the CO pollution. CO emissions from megacities as well as from large fires were investigated applying SICOR, a retrieval algorithm developed for the analysis of newly available TROPOMI data. Results from this work will be relevant to future TROPOMI data analyses.

The manuscript presents valuable work in a mostly clear manner. However, important details of the retrieval process are not explained; they should be clarified in a few words so the reader can better understand the validity of method and results. The manuscript would improve if more effort was placed into the interpretation of results. Grammar needs to be revised. Following are comments listed sequentially. Please double check manuscript for additional grammar issues not included here.

- Page 1/line 1: "upcoming TROPOMI Sentinel-5 Precursor CO data product".

- 1/3: "urban and regional scales [...] from megacities and wildfires, respectively". Regarding the use of the word "megacities": megacities are considered, generally, cities with >= 10 million inhabitants. According to this definition, LA is a megacity. Paris and Tehran are not, unless their metropolitan areas are included. Please clarify.

- 1/6: "observations with low (<1.5 km) and medium-high (<5 km) clouds".

- 1/6 and 3/19: "As an example". Exemplary = "the best of its kind" or "warning, deterrent".

- 1/7: "SCIAMACHY's mean clear-sky observations show weak CO enhancements".

- 1/10-11: Consider Planetary Boundary Layer (PBL) height effects, how they may affect your results. Please see more detailed comments on Section 6.

- 1/13: "information to clear-sky retrievals, which can only be obtained over land".

- 1/18: "will allow improved detection of CO emissions and their vertical extension over cities and fires, making possible new research applications".

- 1/22: "Because of its moderate lifetime and low background concentrations".

- 2/2-5: Please modify (here and elsewhere in the manuscript ) this type of citation list as follows: "Gloudemans et al., 2009; Frankenberg et al., 2005; Buchwitz et al., 2007; Gimeno Garcia et al., 2011".

- 2/7: "The presence of clouds may represent a challenge for the interpretation of satellite observations." Please reword; consider that certain analysis may actually require (or benefit from) the presence of clouds.

- 2/30-33: Please consider moving to Section 2, where details on the datasets utilized or relevant to this study should be presented.

- 2/35-3/10: Same as above

- 3/11: "We have applied the SICOR algorithm to the full SCIAMACHY 2.3 mm dataset."

- 3/14-16: Please consider moving to Section 2, where details on the datasets utilized should be presented. What did those validation efforts find?

- 3/18: "We separate CO retrievals under clear-sky, low (<1.5 km), and medium-high (<5 km) cloud conditions. As an example, we discuss CO pollution over Tehran, Paris, and Los Angeles [. . .] and Alaska/Canada 2004."

- 3/23: Sections 5 and 7 are missing in this list.

- 3/24: Consider modifying title to "SCIAMACHY CO retrievals" or "SCIAMACHY dataset and retrievals", since the section mostly covers the retrieval process. Section 2 lacks a sufficient description of the SCIAMACHY dataset; for example, its spatial resolution is not provided till the end of Section 5. More efforts are put into describing the TROPOMI dataset, even though it is not utilized in this work. Please properly describe the SCIAMACHY dataset in this section.

- 3/26-27: CO data availability is also discussed in section 7; please avoid repetition.

- 3/30: For simplicity: "Auxiliary parameters like signal-to-noise ratio of the measurement and number of retrieval iterations are also provided."

- 3/32: (Here and elsewhere) this type of citation list should read "by Vidot et al. (2012), Borsdorff et al. (2014), Landgraf et al. (2016b)".

- 3/33: Where does the CH4 a priori come from? Is the same a priori used for all locations? for all seasons? for all years?

- 3/33: Please simplify and reword sentence for clarity. Is this what you mean?: "Here, cloud optical depth and height are estimated using prior knowledge about CH4, ECMWF surface pressure, and the observed absorption (of CO? CH4? please clarify) in the spectral fit window."

- 4/3: Please clarify in a few words "the selection of the retrieval window" so the reader understands what that means without reading the cited work. Is that the spectral range used? How was it determined?

- 4/6: For simplicity and clarity, please reword to "In this study we screen SCIAMACHY CO retrievals based on the number of retrieval iterations and the [...]".

- 4/16: Please clarify: 1) are data with clouds above 5 km filtered out? 2) are cloud properties calculated based on a priori CH4 profiles? In a few words: how? 3) are SCIAMACHY radiances corrected or not for cloud scattering effects? for aerosol effects? The later would be particularly relevant when investigating fires 4) how is CO from the "shielded" (by clouds) partial column quantified? Is it ignored? Is it copied/calculated from a priori? If yes, then: What is used as CO a priori? is the same a priori used for all locations? for all seasons? for all years?

- 4/16: "we attribute to erroneous retrievals possibly caused by ..."

- 4/17: "averaging the data over the entire mission period" This was done only for the cities, correct? Please reword accordingly.

- 4/27: if the following is correct, please clarify in the text that low values in the averaging kernel indicate that more of the CO a priori goes into the retrieval. Does SICOR result in averaging kernel values=1 if there is no a priori involved, i.e., if the CO signal is

strong enough to produce a retrieval which is totally independent of the CO a priori?

- 5/2: is reference profile = a priori profile? Please reword for clarity.

- 5/7: "used by Fioletov"

- 5/8-9: "grid cell", "radius", how were they determined? Empirically? Please explain and justify the values used. Are they different for each location analyzed? If yes, then a summary table would be very useful.

- 5/15: references for previous work on Mexico/Guatemala 2005 fires?

- 5/19: "but shifted by about 45 days for both events. We ascribe this temporal shift to the atmospheric response time to built up the high atmospheric CO concentration." Unclear if this is the case. MOPITT data do not seem to show such 45 day shift. Also, CO plumes from Asia cross the Atlantic in just a couple of days. Either support this claim with further evidence or remove from text; additionally, the relevance of this claim to the manuscript's main point is unclear.

- 5/20-22: "As expected, CO retrieval values increase during the fire season (March-May???) each year, coinciding with an increase in burned area."

- 5/30: Please consider swapping the "clear-sky" and "cloudy" order; otherwise, the text seems to imply that cloudy retrievals are the most trustworthy.

- 6/1-26: Please consider organizing text by region for clarity.

- 6/1: "reduced noise in the CO data product" Also: could having more samples reduce the noise?

- 6/4: "This supports our finding that both low-cloud and medium-high cloud retrievals can capture the burning events equally well" Please reword for clarify: this would not be true if the CO plume was confined near the surface, correct?

- 6/7: "0.5 degree" provide in km instead, for consistency.

- 6/10: However, fires elsewhere do not show up in the clear/low cloud CO maps. Please explain.

- 6/19: "can detect the wildfires in agreement with the MODIS burned area product". If one tries to locate fires from the burned area map in the high cloud CO map one finds no coincidences in space. Is the high cloud CO map rather showing CO transported away from the fire areas? Please explain and reword text accordingly.

- 6/21: "more than doubled". That is because now retrievals over both land and water are being obtained; no fires in water-covered regions, though, just CO transported from the fire areas. Please reword to explain this.

- 6/26: Please clarify that what SCIAMACHY "sees" is transported CO away from the fire regions, not the actual fires.

- 6/26: It would be very useful if results from this work were compared to those in Pfister et al. (2005)

- 6/29: "We accumulated all SCIAMACHY observations from 2003 to April 2012 around these cities, distinguishing between clear-sky, low-cloud and medium-high cloud retrievals. Then we applied the oversampling approach with a longitude/latitude grid of $\delta$ = ??? km and an averaging radius r = 40 km (Figs. 6, 7, and 8)." Please move the information on the MODIS urban area contours of Schneider et al. (2009) to figure captions.

- 7/1: " urban area contours (Schneider et al., 2009)".

- 7/5: It seems like the red area is to the southeast of where Rouen would be located.

- 7/8: A similarly red area can be seen southeast of Paris. Please explain. Same north of Rouen. Could it be transported CO?

- 7/12: "Los Angeles"

- 7/12: "Medium-high clouds shield the atmosphere below and so the retrieval is less

sensitive to the city pollution" Does this mean that SICOR retrievals over clouds do not/cannot approximate CO under the clouds? It just uses a priori? Please clarify in Section 2.

- 7/16: Please comment on the influence of PBL heights. A quick look at maps in Engeln and Teixeira (2013) shows that PBL heights for LA and Tehran are quite different from those for Paris. These differences may explain the detectability or not of urban emissions under medium-high cloud conditions. You may also want to discuss seasonal effects on PBL heights and, thus, on urban emissions detectability affected by clouds. (Also: the work presented here does not separate data by season; comment on this.) For example, the PBL height over Tehran and LA in the summer may be » 1.5 km, allowing for city emissions to be detected in some cases under medium-high cloud conditions. In contrast, the PBL height over Paris is approx. <= 1.5 km all year round, thus it may be that no CO signal will be detected if medium-high clouds are present. A simple exercise to test this hypothesis: see if removing summer data in Tehran and LA high cloud maps results in no CO enhancement over these cities.

- 7/17-23: for clarity: summarize in table rather than in text?

- 7/24: Pommier et al (2013) provided CO enhancements over LA, Tehran, and Paris; it would be very helpful if their results were contrasted with results from this work. Even better if their analysis for the 2004-2008 period was replicated with SCIAMACHY data, which is available for this same period.

- 7/26: Please comment on early TROPOMI performance.

- 7/32-8/8: This description of the SCIAMACHY instrument and dataset should be in Section 2.

- 8/10-9/10 for clarity: please consider discussing first cities, then fires (or vice versa) rather than mixing both discussions.

- 8/10: "column retrievals"

- 8/11: "SWIR spectra to study [. . .] complementary [...]"

- 8/12: "SICOR was developed"

- 8/13: "of SWIR measurements [. . .]. SICOR provides the possibility [...]"

- 8/20: "it was even possible to detect pollution over the neighboring city of Rouen". Unclear.

- 8/21: "suitable to locate the source of biomass burning" Clarify that some of the fire regions compiled in the burned area maps were located, but not all of them.

- 8/26: "inferior" Please clarify: coverage was inferior, but not the retrievals themselves, correct?

- 8/28: "Regarding pollution from megacities, the CO results are similar for clear sky and low cloud measurements". This is probably because both sample inside the PBL.

- 8/31: please change to "retrievals [. . .] provide complementary information". Here and elsewhere in the manuscript: please match subject and verb.

- 9/2: "in the SWIR spectral range"

- 9/5: "the retrieval underestimates the total column of CO" TROPOMI's ATBD seems to state the opposite.

- 9/6: "Los Angeles"

- 9/7: Paris also had the lowest delta CO of the three cities; please discuss. For these three cities: can the amount of CO enhancement be traced to population size?

- 9: Are there any controls to make sure CO transported from elsewhere is not being included in the fires analysis? This may be an issue when averaging long periods of time.

- 9: No wind correction (applied in Pommier's work) was applied here. Please justify.

- This should have been discussed early on: why 1.5 and 5 km thresholds were selected?

- 13/fig. 1: Please clarify: is the solid yellow line for clear conditions? If not, include one example.

- 14/fig. 2: Please explain negative CO values in first three panels. For readability, please include monthly markers.

- 15/fig. 3: Why are maps for 2003 not shown?

- 15/fig. 4: Font too small, not legible.

- 16/fig. 5: Is this figure needed?

- 16/fig. 5, 6, and 7: These results are quite remarkable, keeping in mind SCIA-MACHY's spatial resolution. To make this point more clear, please consider adding one panel to each figure with actual SCIAMACHY spatial resolution.

Please clarify what is shown in each of the three panels: clear, low, and high cloud results?

Please remove the latitude/longitude box information in captions since maps have lats and lons.

If scale in fig. 5 was 70-110 the reader could compare better results for the three cities.

Clarify that fig. 6 shows the Paris region.

Font is too small, scale and labels in maps are not legible.

- 17: are fig. 9 and 10 needed?
* * *

---

## Author Comment (AC1) · 6 Apr 2018

**author comments on the manuscript amt-2017-423, referee 1**

We would like to thank the referee for the comments to further improve our manuscript. In this document we provide our reply to the comments. The original comments made by the referee are numbered and typeset in italic and bold face font.

1. ***Generally, the paper discusses an important subject as we lose high percent of data due to cloud screening. However, the paper needs to be more clear as it is ambiguous at many places and also it might be good if the English is edited by professionals.***

   We assume to satisfy the request of the reviewer with our changes to the manuscript.

2. ***The following are examples of the parts that need to be clarified: 1-The authors need to explain why they chose the following oversampling size? The SCIAMACHY data are averaged over an area with a radius r = 90 km and subsequently oversampled with a longitude/latitude sampling distance of = 0.5 degree***

   **adjusted**

   To make it more clear we have added the following paragraph at p5,l10:

   " To find an appropriate averaging radius $r$, a trade-off has to be made between spatial resolution and noise of the averaged CO field. Obviously, this choice depends on the particular application due the number of available CO data points and the brightness of the observed scene. The choice of the sampling distance $\delta$ is less critical as far as it is $< r$ to achieve an oversampling of the data field. Therefore, $r$ and $\delta$ changes for the applications discussed in the following and are provided accordingly in the discussion. "

   Hence, in Sec. 3 we chose $r = 90$ km, and "$\delta = 0.5$" to analyse pollution by wild fires and in Sec. 4 $r = 40$ km, and "$\delta = 0.05$" to analyse pollution by cities.

3. ***2- The following parts need to be rephrased to be more clear Page 2, line 13 Please rephrase to be more clear "Obviously, this assumption does not hold for optically thick water clouds shielding the atmosphere below. Therefore, Buchwitz et al. (2007); Gloudemans et al. (2009); de Laat et al. (2012) used the retrieved CH4 total column and model vertical profiles of CH4 and CO to compensate for the shielding effect by clouds on the estimated total column of CO"***

   **adjusted**

   We have changed the sentence at p2,l13:

   to

   "A different approach is followed by Buchwitz et al. (2007); Gloudemans et al. (2009); de Laat et al. (2012) who use vertical profiles of $CH_4$ and CO taken from model simulations to compensate for the reduced sensitivity when retrieving trace gas columns from cloud contaminated measurements. "

4. ***Page 2, line, 29 Please rephrase to be more clear "This aspect is particularly interesting in the light of upcoming future missions with improved radiometric and spatial sampling performance"***

   **adjusted**

   the sentence at p2,l29 is rephrased

   to

   " On the other hand, cloudy and clear sky observations with different vertical CO sensitivities provide information on the vertical distribution of CO (Liu et al., 2014). Here, it is necessary to observe similar CO vertical distributions with different cloudiness, which will be met more frequently by upcoming CO missions with enhanced spatial sampling and resolution in combination with improved data quality of the individual CO soundings. "

5. ***Page 2, line 30 I think the following reference needs to be updated because it is 2012, and TROOMI launched in 2017. "October 13, 2017 the Tropospheric Monitoring Instrument (TROPOMI) was success- fully launched on the Sentinel-5 Precursor Mission (S-5P) (Veefkind et al., 2012). "?***

   **adjusted** The citation is still valid but we rephrase the sentence to make it clear. We changed the sentence at p2,l30 to

   " October 13th, 2017 the Tropospheric Monitoring Instrument (TROPOMI) was successfully launched on the Sentinel-5 Precursor Mission (S-5P). The mission objectives and requirements are provided by Veefkind et al., 2012. "

6. ***Page 3, line 8 Please rephrase to be more clear light path enhancement***

    **adjusted**

    The sentence at p3,l8 "The TROPOMI CO data product comprises the estimate of the CO column, its noise estimate, effective cloud parameters and the CO column averaging kernel, which reflects the effect of cloud shielding and light path enhancement by clouds on the retrieved CO column. "

    is rephrased to

    "The TROPOMI CO data product comprises the estimate of the CO column, its noise estimate, effective cloud parameters and the CO column averaging kernel, which provides the sensitivity of the retrieved column with respect to changes of the true vertical CO profile. For example for cloudy atmospheres, the averaging kernel reflects the shielding of the atmosphere below the cloud with a reduced CO sensitivity, equivalent to small values of the averaging kernel. "

7. ***Page 9, line 29 Please rephrase to be more clear The following have repeated statements, please delete one of them. "a higher spatial resolution of up to 77 km2 , and improved spatial sampling"***

    **adjusted**

    The sentence at p9,l29 " The 2.3 $\mu$m spectral range of TROPOMI is covered by SCIAMACHY with the same spectral resolution and spectral coverage but TROPOMI is characterized by a significantly improved SNR of the measurements, a higher spatial resolution of up to $7 \times 7$ km$^2$, and improved spatial sampling with daily global coverage. "

    is changed to " The 2.3 $\mu$m spectral range is covered both by TROPOMI and SCIAMACHY with the same spectral resolution, whereby TROPOMI shows an improved radiometric performance with a high spatial resolution of up to $7 \times 7$ km$^2$ and with daily global coverage. "

8. ***Page 8, line, 28 Please rephrase to be more clear For Tehran and Los Angeles the temporal and spatial sampling of low cloud observations improved the spatial coincidence of the CO enhancement and the MODIS urban area product compared to the clear sky CO product.***

    **adjusted**

    We have changed the sentence at p8,l28 to

    "Compared to clear sky observations, the temporal and spatial sampling of low cloud observations improves the spatial match of the CO enhancements with the corresponding MODIS urban areas of Tehran and Los Angeles. "

9. ***Page 9, line, 3 Please rephrase to be more clear "The analysis of the CO pollution from megacities from SCIAMACHY soundings with medium-high clouds indicated a significant reduction in the CO enhancement for the three cities"***

    **adjusted**

    We have rephrased the sentence at p9,l3 to " However, when using medium-high clouds for the detection of CO pollution we recognized a significant reduction in the CO enhancement above the three cities. "

10. ***Page 9, line 14 open op new, replace op with up***

    **adjusted**

11. ***The following figures have very small font so please increase their size: Figures 2,46,7,8***

    **adjusted**

[revised manuscript text omitted]

---

## Author Comment (AC2) · 6 Apr 2018

**author comments on the manuscript amt-2017-423, referee 2**

We would like to thank the referee for the detailed comments to further improve our manuscript. In this document we provide our reply to the comments. The original comments made by the referee are numbered and typeset in italic and bold face font.

1. ***The manuscript presents valuable work in a mostly clear manner. However, important details of the retrieval process are not explained; they should be clarified in a few words so the reader can better understand the validity of method and results.***

   **slightly adjusted**

   In our view, already published work should be cited and only shortly summarized. In this manuscript, we analysis the data product described by Borsdorff et al, 2016, 2017 with respect to CO hot spots due to wildfires and urban pollution. The data set and the specific retrieval settings for the SCIAMACHY CO retrieval are given by Borsdorff et al. 2017 whereas details on the retrieval algorithm are provided by Vidot et al. (2012), Borsdorff et al. (2014), and Landgraf et al. (2016b). All references are provided in the manuscript. We hesitate to discuss these details again and think that the short introduction to the retrieval approach at p2,l32-p3,l5 is sufficient. On the other hand, we realized that information on the spectral fit window may be important in the context of the current manuscript and hence, we have added the sentence:

   " The spectral range for the retrieval from 2311-2338nm was chosen to compensate for the detector pixel loss in the later years of the mission but also to include a strong $CH_4$ absorption line. "

2. ***The manuscript would improve if more effort was placed into the interpretation of results. Grammar needs to be revised. Following are comments listed sequentially. Please double check manuscript for additional grammar issues not included here.***

   **adjusted below**

   We assume to satisfy the request of the reviewer with our changes to the manuscript blow.

3. ***- Page 1/line 1: upcoming TROPOMI Sentinel-5 Precursor CO data product.***

   **adjusted**

4. ***- 1/3: urban and regional scales [. . .] from megacities and wildfires, respectively. Regarding the use of the word megacities: megacities are considered, generally, cities with ¿= 10 million inhabitants. According to this definition, LA is a megacity. Paris and Tehran are not, unless their metropolitan areas are included. Please clarify.***

   **adjusted**

   We renamed megacities to cities throughout the manuscript and adjusted the title accordingly to "Detection of carbon monoxide pollution from cities and wildfires on regional and urban scales: The benefit of CO column retrievals from SCIAMACHY 2.3 $\mu$m measurements under cloudy conditions."

5. ***- 1/6: observations with low (<1.5 km) and medium-high (<5 km) clouds.***

   **adjusted**

   We changed the sentence at p1,l6:

   from ". . . observations and observations with low and medium-high clouds. "

   to ". . . observations and observations with low (<1.5 km) and medium-high clouds (1.5km-5km). "

6. ***- 1/6 and 3/19: As an example. Exemplary = the best of its kind or warning, deter- rent.***

   **adjusted**

   We changed the sentence at p1,l6 and p3,l19:

   from "Exemplary, . . ."

   to "As an example, . . . , "

7. ***- 1/7: SCIAMACHYs mean clear-sky observations show weak CO enhancements.***

   **adjusted**

8. ***- 1/10-11: Consider Planetary Boundary Layer (PBL) height effects, how they may affect your results. Please see more detailed comments on Section 6.***

   **not adjusted** Please see our answer to the comments on Section 6 of this reviewer.

9. **- 1/13: information to clear-sky retrievals, which can only be obtained over land.**

   adjusted

10. **- 1/18: will allow improved detection of CO emissions and their vertical extension over cities and fires, making possible new research applications.**

   adjusted

11. **- 1/22: Because of its moderate lifetime and low background concentrations.**

   adjusted

12. **- 2/2-5: Please modify (here and elsewhere in the manuscript ) this type of citation list as follows: Gloudemans et al., 2009; Frankenberg et al., 2005; Buchwitz et al., 2007; Gimeno Garcia et al., 2011.**

   adjusted

13. **- 2/7: The presence of clouds may represent a challenge for the interpretation of satellite observations. Please reword; consider that certain analysis may actually require (or benefit from) the presence of clouds.**

   adjusted

   We have changed the sentence at p2,l7:

   from "For the interpretation of satellite observations, the presence of clouds in the observed scene represents a major challenge."

   to " The presence of clouds may represent a challenge for remote sensing of CO from space."

14. **- 2/30-33: Please consider moving to Section 2, where details on the datasets utilized or relevant to this study should be presented. - 2/35-3/10: Same as above**

   **not adjusted**

   We carefully considered the reviewer's comment to move parts of the introduction to Section 2 but came to the conclusion that this would weaken the introduction of the manuscript. In our view, the introduction must give an overview on recent research on the subject of the manuscript, also in context of upcoming challenges in the research field. Therefore, we are convinced that it is needed to mention TROPOMI as the successor of SCIAMACHY in the introduction. For example, it is otherwise hard to understand why we apply the TROPOMI operational retrieval code on the SCIAMACHY data. Because we view this work also in the perspective of the ongoing TROPOMI data processing, it is desirable to mention this aspect in the introductory part of the manuscript.

15. **- 3/11: We have applied the SICOR algorithm to the full SCIAMACHY 2.3 mm dataset.**

   adjusted

   To make it more clear we have changed the sentence at p3,l11:

   from "To evaluate the benefit and maturity of this approach for the detection of localized CO enhancements by TROPOMI we have applied the SICOR algorithm to the full SCIAMACHY mission data set of 2.3 $\mu$m spectra. "

   to "We have applied the SICOR algorithm to the full SCIAMACHY 2.3 $\mu$m dataset."

16. **- 3/14-16: Please consider moving to Section 2, where details on the datasets utilized should be presented. What did those validation efforts find?**

   **adjusted** The manuscript uses the SCIAMACHY data set described in the previous work (Borsdorff et al., 2016 and 2017), which includes an extensive validation of the data product with TCCON/NDACC ground-based observation as well as IAGOS airborne measurements. To our opinion, this belongs to the introduction of the manuscript, in line with our view explained above (reply to comment 14). We agree that the findings of the main validation should be mentioned here and therefore we add the following sentence at p3, l16:

   " In general, those studies found a good agreement with the validation datasets considering the high noise error of the SCIAMACHY CO dataset. For most sites the bias is < 10 ppb but can increase significantly at CO hot spots due to representation errors of the validation. "

17. **- 3/18: We separate CO retrievals under clear-sky, low (<1.5 km), and medium-high (<5 km) cloud conditions. As an example, we discuss CO pollution over Tehran, Paris, and Los Angeles [. . .] and Alaska/Canada 2004.**

**adjusted**

We follow the editor's suggestion and have changed the sentence at p3,l18:

from "Here, we distinguish CO retrievals from measurements under clear-sky, low and medium-high cloud conditions. As an example, we discuss the CO pollution by Tehran, Paris, and Los Angeles as well as the wildfires in Mexico/Guatemala 2005 and Alaska/Canada 2014. "

to

"We separate CO retrievals under clear-sky, low (<1.5 km), and medium-high (1.5-5 km) cloud conditions. As an example, we discuss CO pollution over Tehran, Paris, and Los Angeles as well as the wildfires in Mexico/Guatemala 2005 and Alaska/Canada 2004. "

18. **- 3/23: Sections 5 and 7 are missing in this list.**

**adjusted**

We corrected this and adjusted the sentence at p3,l23:

from "The paper is structured as follows: In section 1, we present the SCIAMACHY CO dataset. Section 2 analyzes the benefit of the SCIAMACHY CO retrievals under cloudy conditions to detect wildfires, and Section 3 focuses on the CO emission from megacities. Finally, Section 4 will give a summary and conclusions. "

to " The paper is structured as follows: In section 2, we present the SCIAMACHY CO dataset. Section 3 analyzes the benefit of the SCIAMACHY CO retrievals under cloudy conditions to detect wildfires, and Section 4 focuses on the CO emission from megacities. Section 5 draws some conclusions on the upcoming TROPOMI CO dataset. The summary and conclusion is given in section 6 and finally, section 7 states the availability of the data. "

19. **- 3/24: Consider modifying title to SCIAMACHY CO retrievals or SCIAMACHY dataset and retrievals, since the section mostly covers the retrieval process. Section 2 lacks a sufficient description of the SCIAMACHY dataset; for example, its spatial resolution is not provided till the end of Section 5. More efforts are put into describing the TROPOMI dataset, even though it is not utilized in this work. Please properly describe the SCIAMACHY dataset in this section.**

**adjusted**

We change the title of the section to "SCIAMACHY CO dataset and retrievals"

Furthermore, we add the following paragraph p3, l24:

" The SCIAMACHY instrument was operational on ESA's ENVISAT satellite from January 2003 until April 2012. We utilize the SWIR measurements of SCIAMACHY in nadir observation geometry with a spatial resolution of about 120x30 km$^2$, a swath of 960, and a global coverage within 3 days (Bovensmann et al. 1999).
"

20. **- 3/26-27: CO data availability is also discussed in section 7; please avoid repetition.**

**adjusted**

Correct, we have changed the sentence at p3,l26:

from "The CO data product is available on the public ftp site `ftp://ftp.sron.nl/pub/pub/DataProducts/SCIAMACHY_CO/`. It consists of . . . "

to "The CO data product consists of . . . "

21. **- 3/30: For simplicity: Auxiliary parameters like signal-to-noise ratio of the measurement and number of retrieval iterations are also provided.**

**adjusted**

The sentence at p3,l30 is adjusted

from "Moreover, different auxiliary parameters are provided like the number of iterations of the inversion ($N_{iter}$) and the maximum signal to noise ratio of the measurement SNR$_{max}$.
"

to "Auxiliary parameters like signal-to-noise ratio $SNR_{max}$ of the measurement and number of retrieval iterations ($N_{iter}$) are also provided. "

22. *- 3/32: (Here and elsewhere) this type of citation list should read by Vidot et al. (2012), Borsdorff et al. (2014), Landgraf et al. (2016b).*

   adjusted

23. *- 3/33: Where does the CH4 a priori come from? Is the same a priori used for all locations? for all seasons? for all years?*

   adjusted

   We add the following sentence at p3,l33:

   "The $CH_4$ data was taken from a TM5 model run (Williams et al. 2013, Williams et al. 2014) spanning the entire mission period of SCIAMACHY with global $3\times2$ degree$^2$ horizontal resolution and 3 hourly sampling time."

24. *- 3/33: Please simplify and reword sentence for clarity. Is this what you mean?: Here, cloud optical depth and height are estimated using prior knowledge about CH4, ECMWF surface pressure, and the observed absorption (of CO? CH4? please clarify) in the spectral fit window.*

   adjusted

   To make it more clear we changed the sentence at p3,l33:

   from " Here, cloud optical depth and height are estimated using prior knowledge about $CH_4$, ECMWF surface pressure, and the observed absorption in the spectral fit window."

   to "Here, cloud optical depth and height are estimated using prior knowledge about $CH_4$, ECMWF surface pressure, and the observed $CH_4$ absorption in the 2.3 $\mu$m spectral fit window (Landgraf et al., 2016)."

25. *- 4/3: Please clarify in a few words the selection of the retrieval window so the reader understands what that means without reading the cited work. Is that the spectral range used? How was it determined?*

   adjusted

   To make it more clear we add the following sentence at p4,l3:

   " The spectral range for the retrieval from 2311-2338nm was chosen to compensate for the detector pixel loss in the later years of the mission but also to include a strong $CH_4$ absorption line, which are beneficial for the retrieval of the effective cloud parameters. "

26. *- 4/6: For simplicity and clarity, please reword to In this study we screen SCIAMACHY CO retrievals based on the number of retrieval iterations and the [...].*

   adjusted

   changed the sentence at p4,l6:

   from "In this study, we apply a data screening of the individual SCIAMACHY CO retrievals based on the number of iterations $N_{iter}$ and the . . ."

   to " In this study, we filter the SCIAMACHY CO data based on the number of iterations $N_{iter}$ and the . . ."

27. *- 4/16: Please clarify: 1) are data with clouds above 5 km filtered out? 2) are cloud properties calculated based on a priori CH4 profiles? In a few words: how? 3) are SCIAMACHY radiances corrected or not for cloud scattering effects? for aerosol effects? The later would be particularly relevant when investigating fires 4) how is CO from the shielded (by clouds) partial column quantified? Is it ignored? Is it copied/calculated from a priori? If yes, then: What is used as CO a priori? is the same a priori used for all locations? for all seasons? for all years?*

   1.) adjusted We change the sentence at page p4, l18:

   from

   " To distinguish the effect of clouds on the retrieval, we consider three different categories of cloudy observations as indicated in Tab. 2.
   "

to

" In this study we consider clear-sky and cloudy-sky retrievals with cloud centre heights smaller than 5 km, latter distinguished in three categories as specified in Tab. 2.
"

2.) **not adjusted** This is described in the manuscript p3, l33 " "Here, cloud optical depth and height are estimated using prior knowledge about $CH_4$, ECMWF surface pressure, and the observed $CH_4$ absorption in the 2.3 $\mu$m spectral fit window. (Landgraf et al., 2016) "

3.) **not adjusted** No, we do not apply any cloud clearing to the measurement. This is also not mentioned in the text or any given reference of the SICOR algorithm.

4.) **adjusted** Priori CO information comes from the TM5 model as stated in our reply to the reviewer's comment 23. The retrieval approach is based on the profile scaling approach discussed in Borsdorff et al. 2011 and referenced in the paper. We have summarized its main feature in the manuscript at p4, l29:

"Because the CO column is estimated by a scaling of a reference profile, CO variations above a cloud also induce an adjustment of the CO concentration below the cloud, where the measurement is not sensitive to. "

28. *- 4/16: we attribute to erroneous retrievals possibly caused by ...*

    **adjusted**

    We have changed the sentence at p4,l16:

    from "This filter removes outliers of our data sets, which we attribute to erroneous retrievals rather than an atmospheric signal in case of the selected megacities. "

    to " This filter removes outliers of our data sets, which we attribute to erroneous retrievals possibly caused by the instrument degradation rather than an atmospheric signal in case of the selected megacities. "

29. *- 4/17: averaging the data over the entire mission period This was done only for the cities, correct? Please reword accordingly.*

    **adjusted**

    We have changed the sentence at p4,l17:

    from "It enables us to detect the relatively weak CO enhancement after averaging the data over the entire mission period. "

    to

    "It enables us to detect the relatively weak CO enhancement above cities after averaging the data over the entire mission period. "

30. *- 4/27: if the following is correct, please clarify in the text that low values in the averaging kernel indicate that more of the CO a priori goes into the retrieval. Does SICOR result in averaging kernel values=1 if there is no a priori involved, i.e., if the CO signal is strong enough to produce a retrieval which is totally independent of the CO a priori?*

    **not adjusted**

    The definition of the total column averaging including the meaning of values of 1, higher and lower than 1 are discussed in detail in the manuscript. Please see p4,l20 - p5,l3.

31. *- 5/2: is reference profile = a priori profile? Please reword for clarity.*

    **not adjusted**

    In the manuscript we avoid the term a priori profile because it refers often to a priori information of statistical regularization schemes like the Optimal Estimation Method. The reference profile is defined as the profile scaled during retrieval (see the manuscript p4, l28). Hence, there is no conflict of definition and the term reference profile is also used in our previous works.

32. *- 5/7: used by Fioletov*

    **adjusted**

33. *- 5/8-9: grid cell, radius, how were they determined? Empirically? Please explain and justify the values used. Are they different for each location analyzed? If yes, then a summary table would be very useful.*

    **adjusted**

To make it more clear we add the following paragraph at p5,l10:

" To find an appropriate averaging radius $r$, a trade-off has to be made between spatial resolution and noise of the averaged CO field. Obviously, this choice depends on the particular application due the number of available CO data points and the brightness of the observed scene. The choice of the sampling distance $\delta$ is less critical as far as it is $< r$ to achieve an oversampling of the data field. Therefore, $r$ and $\delta$ changes for the applications discussed in the following and are provided accordingly in the discussion. "

Hence, in Sec. 3 we chose $r = 90$ km, and "$\delta = 0.5$" to analyse pollution by wild fires and in Sec. 4 $r = 40$ km, and "$\delta = 0.05$" to analyse pollution by cities.

34. *- 5/15: references for previous work on Mexico/Guatemala 2005 fires?*

    **adjusted**

    We have added the reference from Herrera et al. 2016 about "Mexican forest fires and their decadal variations" at p5, l13.

35. *- 5/19: but shifted by about 45 days for both events. We ascribe this temporal shift to the atmospheric response time to built up the high atmospheric CO concentration. Unclear if this is the case. MOPITT data do not seem to show such 45 day shift. Also, CO plumes from Asia cross the Atlantic in just a couple of days. Either support this claim with further evidence or remove from text; additionally, the relevance of this claim to the manuscripts main point is unclear.*

    **adjusted**

    We agree with the reviewer's criticism and have changed the sentence p5, l19 form:

    " We ascribe this temporal shift to the atmospheric response time to built up the high atmospheric CO concentration. "

    to

    "The reason for the shift is unclear and will be studied in future also looking at other satellite observations."

36. *- 5/20-22: As expected, CO retrieval values increase during the fire season (March-May???) each year, coinciding with an increase in burned area.*

    **adjusted**

    We have adapted the sentence at p5,l20-22:

    from " It is interesting to note that the retrieval shows both a slowly varying increase and decrease of the burning activities over the month and enhanced peak events, both also evident in the GFED4 Burned Area data. "

    to " As expected, CO retrieval values increase during the fire season (March- May) each year, coinciding with an increase in burned area. Here, the peak events are evident in both the low cloud and medium-high cloud data records. "

37. *- 5/30: Please consider swapping the clear-sky and cloudy order; otherwise, the text seems to imply that cloudy retrievals are the most trustworthy.*

    **adjusted**

    We changed the sentence at p5,l30:

    from "However, the good correlation coefficient and the low bias shows that within the noise limitation the clear-sky retrievals are in good agreement with the cloudy retrievals. "

    to " However, the good correlation coefficient and the low bias shows that within the noise limitation the cloudy retrievals are in good agreement with the clear-sky retrievals. "

38. *- 6/1-26: Please consider organizing text by region for clarity.*

    **not adjusted**

    We want to stress the performance of our retrieval under different cloud conditions. Therefore we chose to organize the text by clear-sky, low cloud, and medium high-clouds atmospheric conditions. This important aspect for us would be lost when reorganizing the text as suggested.

39. *- 6/1: reduced noise in the CO data product Also: could having more samples reduce the noise?*

    **not adjusted**

Yes, the error of the mean would reduce when averaging single measurements. However, for the observation of wildfires temporal averaging is not an option because the events occur occasionally. Therefore, the fact that clear sky observations of wildfires compared to cloudy observations are more noisy cannot be solved by averaging more data.

40. **- 6/4: This supports our finding that both low-cloud and medium-high cloud retrievals can capture the burning events equally well Please reword for clarify: this would not be true if the CO plume was confined near the surface, correct?**

    **adjusted**

    To make it more clear we changed the sentence at p6,l4:

    from "This supports our finding that both low-cloud and medium-high cloud retrievals can capture the burning events equally well, something one may expect since CO pollution from wild fires constitutes a strong source that can reach the free troposphere (see e.g. Yurganov et al.(2005)). "

    to

    "This supports our finding that both low-cloud and medium-high cloud retrievals can capture the burning events equally well, something one may expect since CO pollution from wild fires constitutes a strong source that can reach the free troposphere (see e.g. Yurganov et al. (2005)). In case the CO plume was confined to the near-surface atmosphere, it would be more difficult if not impossible to sense it with cloudy observations. "

41. **- 6/7: "0.5 degree" provide in km instead, for consistency.**

    **adjusted**

    We understand the reviewer request to provide both, the averaging radius $r$ and the sampling distance $\delta$ in km. However, it is common and to our opinion good practice to provide satellite data on a longitude/latitude grid. Our philosophy of data reduction is to stick to this representation as much as possible, which explains that the sampling distance $\delta$ is still given in degree. To make the smoothing comparable for the different cases, we choose for a smoothing radius in km. As explained in the revised version of the manuscript (px ly), the sampling is less critical as far as the smoothed CO fields oversampled. Obviously, the transformation of the sampling in km depends on latitude and to comply with the reviewers concern, we added to text the maximum sampling distance of 55 km at the equator. So, we changed the text at p6 l7 from "0.5 degree" to "0.5 degree ($\leq 55$ km)".

42. **- 6/10: However, fires elsewhere do not show up in the clear/low cloud CO maps. Please explain.**

    **adjusted**

    We add the following sentence at p6, l10:

    " Some fires shown by the MODIS data are not reflected by the SCIAMACHY CO data, which may be explained by the fact that the CO emission of these fires is not sufficient to become detectable with SCIAMACHY observations. "

43. **- 6/19: can detect the wildfires in agreement with the MODIS burned area product. If one tries to locate fires from the burned area map in the high cloud CO map one finds no coincidences in space. Is the high cloud CO map rather showing CO transported away from the fire areas? Please explain and reword text accordingly.**

    **adjusted**

    The reviewer's suggestion is plausible and we have adjusted the text accordingly. This, we changed the sentence at p6,l19 from

    "For medium-high clouds, the corresponding CO product shows much better coverage and so can detect the wildfires in agreement with the MODIS burned area product. "

    to

    " For medium-high clouds, the corresponding CO product shows much better coverage and so can detect enhanced CO concentration transported away from the fires as indicated by the MODIS burned area product. "

44. **- 6/21: more than doubled. That is because now retrievals over both land and water are being obtained; no fires in water-covered regions, though, just CO transported from the fire areas. Please reword to explain this.**

**adjusted**

To make it more clear we changed the sentence at p6,l20:

from "The benefit of cloudy observations to detect wildfires becomes also clear when comparing the number of individual SCIAMACHY CO sounding in Fig. 6. For the Mexico fires, the 2402 individual clear sky soundings are more than doubled (6126 soundings) when we consider low cloud observations"

to "The benefit of cloudy observations to detect the transport of enhanced CO concentration from wildfires becomes also clear when comparing the number of individual SCIAMACHY CO sounding in Fig. 6. For the considered area of the Mexico fires, the 2402 individual clear sky soundings are more than doubled (6126 soundings) when we consider low cloud observations, partly due to additional soundings over ocean which cannot be exploited for clear sky conditions. "

45. *- 6/26: Please clarify that what SCIAMACHY sees is transported CO away from the fire regions, not the actual fires.*

**adjusted**

we change the the sentence at p6,l26:

from

" Due to this, the means to observe wildfire with SCIAMACHY is clearly improved by using cloudy observations in addition to clear sky observations. "

to

" Due to this, the means to observe enhanced CO values by pollution transport from wildfire with SCIA-MACHY is clearly improved by using cloudy observations in addition to clear sky observations.
"

46. *- 6/26: It would be very useful if results from this work were compared to those in Pfister et al. (2005)*

**adjusted**

The results of Pfister are more related to estimating the source strength of the burning what is difficult to compare with our measurements. However, it is definitely valuable to refer to this work and so we add the following sentence at p6,l19:

"In particular, this finding agrees with the study by Pfister et al. (2004) who reported enhanced CO concentration even high up in the atmosphere at about 400 hPa due to the Alaska fires."

47. *- 6/29:"We accumulated all SCIAMACHY observations from 2003 to April 2012 around these cities, distinguishing between clear-sky, low-cloud and medium-high cloud retrievals. Then we applied the oversampling approach with a longitude/latitude grid of = ??? km and an averaging radius r = 40 km (Figs. 6, 7, and 8)" Please move the information on the MODIS urban area contours of Schneider et al. (2009) to figure captions.*

**adjusted**

We changed the paragraph:

from " Accumulating all SCIAMACHY observations from 2003 to April 2012 around these cities and distinguishing between clear-sky, low-cloud and medium-high cloud retrievals, we apply the oversampling approach with a longitude/latitude grid of $\delta = 0.05$ degree and an averaging radius $r = 40$ km as shown in the Figs. 6, 7, and 8 together with the MODIS urban area contours of Schneider (2009). "

to " We accumulated all SCIAMACHY observations from 2003 to April 2012 around these cities, distinguishing between clear-sky, low-cloud and medium-high cloud retrievals. Then we applied the oversampling approach with a longitude/latitude grid of $\delta = 0.05$ degree ($\leq 5.5$ km) and an averaging radius $r = 40$ km (Figs. 6, 7, and 8). "

and add the following sentence to the caption of Fig. 6:

" The urban area contours are based on MODIS measurements (Schneider at al. (2009)). "

48. *- 7/1: urban area contours (Schneider et al., 2009).*

**adjusted**

49. *- 7/5: It seems like the red area is to the southeast of where Rouen would be located. - 7/8: A similarly red area can be seen southeast of Paris. Please explain. Same north of Rouen. Could it be transported CO?*

**adjusted**

The reviewer is right that in this study we cannot conclude on the origin of the enhance CO values. Therefore, we change the sentence p5,l5 from:

"Furthermore, in case of Paris we can detect enhanced CO levels over the neighboring city Rouen (see Fig. 6). "

to

"Furthermore, in case of Paris we detect enhanced CO levels near the neighboring city Rouen caused by local emissions or transport from the remote pollution of Paris (see Fig. 6). "

50. *- 7/12: Los Angeles*

    **adjusted**

51. *- 7/12: Medium-high clouds shield the atmosphere below and so the retrieval is less sensitive to the city pollution Does this mean that SICOR retrievals over clouds do not/cannot approximate CO under the clouds? It just uses a priori? Please clarify in Section 2.*

    **not adjusted**

    This is already explained in section 2 p4,l28-30:

    " Because the CO column is estimated by a scaling of a reference profile, CO variations above a cloud also induce an adjustment of the CO concentration below the cloud, where the measurement is not sensitive to. "

    This is a characteristic of the profile scaling approach.

52. *- 7/16: Please comment on the influence of PBL heights. A quick look at maps in Engeln and Teixeira (2013) shows that PBL heights for LA and Tehran are quite different from those for Paris. These differences may explain the detectability or not of urban emissions under medium-high cloud conditions. You may also want to discuss seasonal effects on PBL heights and, thus, on urban emissions detectability affected by clouds. (Also: the work presented here does not separate data by season; comment on this.) For example, the PBL height over Tehran and LA in the summer may be 1.5 km, allowing for city emissions to be detected in some cases under medium-high cloud conditions. In contrast, the PBL height over Paris is approx. <= 1.5 km all year round, thus it may be that no CO signal will be detected if medium-high clouds are present. A simple exercise to test this hypothesis: see if removing summer data in Tehran and LA high cloud maps results in no CO enhancement over these cities.*

    **not adjusted**

    Calculating temporal subsets of CO enhancements of city scales from SCIAMACHY observation is too ambitious and is not an option considering the noise level of the SCIAMACHY CO retrievals. To distinguish the pollution above the cities from the background, we need the full 9 years of measurements. The point raised by the reviewer is very interesting and hopefully can be studied with the new TROPOMI CO data set.

53. *- 7/17-23: for clarity: summarize in table rather than in text?*

    **adjusted**

    we added a table with the numbers and changed the text p7,l17-23 from:

    " Including cloud contaminated soundings means about double the amount of data is available for Tehran (a factor of 2.1) and Paris (a factor of 2.6) and Los Angeles (a factor of 1.8). The relative amount of cloud contaminated measurements differs significantly per city. For Tehran, we obtain 47 % (2674) clear-sky and 44 % (2501) and 9 % (537) low and medium-high cloud observations, respectively. A similar distribution holds for Los Angeles with 55 % (2557) clear-sky and 35 % (1630) and 10 % (482) soundings for low and medium cloud conditions, whereas for Paris the situation differs with 38 % (1338) clear-sky soundings and 22 % (766) and 40 % (1388) low and medium cloud soundings.
    "

    to

    " Including cloud contaminated soundings means about double the amount of data is available for Tehran (a factor of 2.1) and Paris (a factor of 2.6) and Los Angeles (a factor of 1.8). The relative amount of cloud contaminated measurements differs significantly per city and is summaries in Tab. 2. Tehran and Los Angeles show a similar distribution with a high number of clear-sky and low cloud observations, whereas for Paris cloudy measurements are predominant. "

54. *- 7/24: Pommier et al (2013) provided CO enhancements over LA, Tehran, and Paris; it would be very helpful if their results were contrasted with results from this work. Even better if their analysis for the 2004-2008 period was replicated with SCIAMACHY data, which is available for this same period.*

   **not adjusted**

   We understand that this would be desirable however the SCIAMACHY dataset needs the full mission range to sense the enhancements above the cities as stated above. Furthermore, a direct contrast with the study of Pommier is not possible because our method is different.

55. *- 7/26: Please comment on early TROPOMI performance.*

   **adjusted**

   A publication with first results about the TROPOMI CO dataset are already published. Accordingly, we will add the following sentence at p8,l8:

   " First results of the TROPOMI CO dataset are reported in Borsdorff et al. (2018). " Since other TROPOMI publications are in preparation we don't want to include these here.

56. *- 7/32-8/8: This description of the SCIAMACHY instrument and dataset should be in Section 2.*

   **not adjusted**

   Our intention is to give an outlook of the TROPOMI instrument and we belief that the placement of the text at the end of the manuscript is logical.

57. *- 8/10-9/10 for clarity: please consider discussing first cities, then fires (or vice versa) rather than mixing both discussions.*

   **not adjusted**

   The structure suggested by the reviewer was actually our first attempt. However, since we want stress the difference between clear-sky and cloudy-sky retrievals and their quality, we decided to structure the section in the way it is presented, which we think is still justified.

58. *- 8/10: column retrievals*

   **adjusted**

59. *- 8/11: SWIR spectra to study [. . .] complementary [...]*

   **adjusted**

60. *- 8/12: SICOR was developed*

   **adjusted**

61. *- 8/13: of SWIR measurements [. . .]. SICOR provides the possibility [...]*

   **adjusted**

62. *- 8/20: it was even possible to detect pollution over the neighboring city of Rouen. Unclear.*

   **adjusted**

   we changed the sentence at p8,l20:

   from "For Paris, it was even possible to detect pollution over the neighboring city of Rouen."

   to "For Paris, we detected enhanced CO values next to the neighboring city of Rouen, caused by the city itself or transport of remote pollution of Paris."

63. *- 8/21: suitable to locate the source of biomass burning Clarify that some of the fire regions compiled in the burned area maps were located, but not all of them.*

   **adjusted**

   To make it more clear we changed the sentence at p8,l21:

   from "Furthermore, clear-sky retrievals turned out to be suitable to locate the source of biomass burning in Mexico/Guatemala in agreement with the daily GFED4 Burned Area data product. "

   to "Furthermore, clear-sky retrievals turned out to be suitable to locate the source of biomass burning in Mexico/Guatemala in agreement with most of the burned area reported by the daily GFED4 product. "

64. **- 8/26: inferior Please clarify: coverage was inferior, but not the retrievals themselves, correct?**

**adjusted**

The quality of clear-sky retrievals strongly depends on the ground reflectivity. By that, not only the coverage was inferior to retrievals under cloudy conditions but also the retrieval noise performance considering that clouds are highly reflected in the SWIR.

To make it more clear we changed the sentence at p8,l26:

from " We found clear-sky retrievals for the wild fires in Mexico/Guatemala 2005 inferior to cloudy retrievals regarding the noise performance and temporal and spatial sampling. "

to " We found clear-sky retrievals for the wild fires in Mexico/Guatemala 2005 inferior to cloudy retrievals regarding the noise performance (clouds are high reflectivity in the SWIR), and the temporal and spatial sampling.
"

65. **- 8/28: Regarding pollution from megacities, the CO results are similar for clear sky and low cloud measurements. This is probably because both sample inside the PBL.**

**adjusted**

we add the sentence at p8,l28:

" This is probably because both are sensitive to CO in the planetary boundary layer."

66. **- 8/31: please change to retrievals [. . .] provide complementary information. Here and elsewhere in the manuscript: please match subject and verb.**

**adjusted**

67. **- 9/2: in the SWIR spectral range**

**adjusted**

68. **- 9/5: the retrieval underestimates the total column of CO TROPOMIs ATBD seems to state the opposite.**

**adjusted**

This is not necessarilly a contradiction. The retrieved CO column can over or underestimated the true column depending on the discrepancy between the reference profile, scaled during the retrieval, and the true vertical CO profile. In our case, the reference profile does not include features of urban CO pollution because of the coarse spatial resolution of the TM5 model ($2 \times 3$ degree). Hence, when the satellite becomes insensitive for the lower atmosphere because of cloud contamination, the retrieved CO column underestimates the true column. If the reference profile would overestimate the CO gradient caused by the pollution of cities it would be the other way round.

We added the following sentence at p5, l3

" Hence, depending on this discrepancy the retrieved column can over or underestimate the true vertical column.
"

69. **- 9/6: Los Angeles**

**adjusted**

70. **- 9/7: Paris also had the lowest delta CO of the three cities; please discuss. For these three cities: can the amount of CO enhancement be traced to population size?**

**not adjusted**

The link between CO enhancement and population size goes much beyond the scope of this manuscript. In first instance, we expect the CO enhancement to be correlated with surface emissions. However, to make here any statement, regional model simulations are needed. We do not want to speculate and, to our opinion, this should be a subject for future research.

71. **- 9: Are there any controls to make sure CO transported from elsewhere is not being included in the fires analysis? This may be an issue when averaging long periods of time.**

**not adjusted**

No control mechanism is implemented yet. The reviewer is rising an interesting point, that could be addressed with an extensive atmospheric modeling approach. However, this is out of scope for this study.

72. **- 9: No wind correction (applied in Pommiers work) was applied here. Please justify. - This should have been discussed early on: why 1.5 and 5 km thresholds were selected?**

**not changed**

We tried the wind correction suggested by Pommier. However, we have not seen an improvement to the more simpler approach we are following. The reason could be the spatial resolution of SCIAMACHY. However, we believe that for total column measurements of CO it is difficult if not impossible to assign a representative wind direction. Therefore, this is not discussed in more detail in the manuscript.

73. **- 13/fig. 1: Please clarify: is the solid yellow line for clear conditions? If not, include one example.**

**adjusted**

We add the following sentence to the caption of Fig.1:

"Here, the solid yellow line represent clear-sky conditions."

74. **- 14/fig. 2: Please explain negative CO values in first three panels. For readability, please include monthly markers.**

**adjusted**

We add the following sentence at px,ly:

"Depending on the signal-to-noise ratio of the measurements the retrieval noise of individual CO retrieval can exceed 100 % of the retrieved column and by that can result in negative CO columns. It is important not to reject negative values when averaging data to avoid artificial biases (de Laat et al (2007), Gloudemanns et al. (2009))."

75. **- 15/fig. 3: Why are maps for 2003 not shown?**

**adjusted**

We added an additional Figure for the burning in Mexico 2003. Respectively the number of the Figure will be 4. Furthermore we changed the sentence at p6,l12:

from " Also, the earlier burning event in Mexico 2003 in Fig. 2 followed a similar transport pattern of enhanced CO over the oceans (not shown). "

to "Also, the earlier burning event in Mexico 2003 in Fig. 2 followed a similar transport pattern of enhanced CO over the oceans (as shown in Fig. 4). "

76. **- 15/fig. 4: Font too small, not legible.**

**adjusted**

We enlarged the Font size in Fig. 4.

77. **- 16/fig. 5: Is this figure needed?**

**not adjusted**

For the manuscript, it is important to show that the statistics of measurements under the different cloud contamination cases differ between the Mexico and Alaska case.

78. **- 16/fig. 5, 6, and 7: These results are quite remarkable, keeping in mind SCIAMACHYs spatial resolution. To make this point more clear, please consider adding one panel to each figure with actual SCIAMACHY spatial resolution. Please clarify what is shown in each of the three panels: clear, low, and high cloud results? Please remove the latitude/longitude box information in captions since maps have lats and lons. If scale in fig. 5 was 70-110 the reader could compare better results for the three cities. Clarify that fig. 6 shows the Paris region. Font is too small, scale and labels in maps are not legible.**

**adjusted**

We changed the caption of Fig. 6

from "SCIAMACHY CO column mixing ratio averaged from January 2003 to April 2012 in the latitude/longitude box [(49.9°N,0.7°E), (47.8°N,4.0°E)]. The resolution of the plot is 0.05 degree in latitude and longitude and the data is oversampled using a radius of 40km. "

to "SCIAMACHY CO column mixing ratio averaged from January 2003 to April 2012 under clear-sky (left panel), low cloud (middle panel), and medium-high cloud (right panel) atmospheric conditions above Paris. The spatial sampling of the plot is $\delta = 0.05$ degree in latitude and longitude and the data are averaged with radius $r = 40$ km. "

We changed the caption of Fig. 7

from "Same as Fig. 6 but for Tehran using the latitude/longitude box [(36.6°N,49.7°E), (34.3°N,52.5°E)]."

to

"Same as Fig. 6 but for Tehran."

We changed the caption of Fig. 8

from " Same as Fig. 6 but for Los Angeles using the latitude/longitude box [(35.2°N,119.6°W), (32.9°N,116.9°W)]." to " Same as Fig.6 but for Los Angeles."

Furthermore, we enlarged the font of the x,y labels, the labels of the color bar of figure 6,7,8. For consistency, we also removed the latitude/longitude box information inf Fig. 3 and Fig. 4. However, we don't think that an extra panel should be added to illustrate the resolution of the SCIAMACHY instrument. For Paris, the chosen color scale of Fig. 6 is need since the background concentration of CO differs between Paris, Los Angeles, and Tehran and additionally the enhancement of CO above Paris is much lower than for the other cities.

79. *- 17: are fig. 9 and 10 needed?*

**not adjusted**

Figure 9 illustrates the statistics of measurements under different cloud contaminations and shows that it depends strongly on the considered location. Figure 10 quantifies the CO enhancement with respect to the background signal and so summarizes the capability to detect CO pollutions for the different data sets. We are convinced that both figures are important to support the main conclusions of the manuscript.

[revised manuscript text omitted]